### Northern Hemisphere Surface Freeze/Thaw Product from Aquarius L-band Radiometers

Michael Prince<sup>1,2</sup>, Alexandre Roy<sup>3,2,1</sup>, Ludovic Brucker<sup>4,5</sup>, Alain Royer<sup>1,2</sup>, Youngwook Kim<sup>6</sup>, Tianjie Zhao<sup>7</sup>

5

<sup>1</sup> Centre d'Applications et de Recherches en Télédétection (CARTEL), Université de Sherbrooke, Sherbrooke, QC J1K 2R1, Canada

<sup>2</sup> Centre d'Étude Nordique, Université Laval, Québec, Canada

<sup>3</sup> Université de Montréal, Département de Géographie, Montréal, QC, H2B 2V8, Canada

<sup>4</sup>NASA Goddard Space Flight Center, Cryospheric Sciences Laboratory, Code 615, Greenbelt, MD 20771, USA

<sup>5</sup> Universities Space Research Association, Goddard Earth Sciences Technology and Research Studies and investigations, Columbia, MD 21044, USA

<sup>6</sup>Numerical Terradynamic Simulation Group, College of Forestry & Conservation, The University of Montana, Missoula, MT 59812, USA

<sup>7</sup> State Key Laboratory of Remote Sensing Science, Institute of Remote Sensing and Digital Earth, Chinese Academy of Sciences, Beijing, China

Correspondence to: Michael Prince (michael.prince@usherbrooke.ca)

**Abstract.** In the Northern Hemisphere, seasonal changes in surface freeze/thaw (FT) cycle are an important component of surface energy, hydrological and eco-biogeochemical processes that must be accurately monitored. This paper presents the

- weekly polar-gridded Aquarius passive L-Band surface freeze/thaw product (FT-AP) distributed on the Equal-Area Scalable Earth Grid version 2.0, above the parallel 50° N, with a spatial resolution of 36 km x 36 km. The FT-AP classification algorith m is based on a seasonal threshold approach using the normalized polarization ration, references for frozen and thawed conditions and optimized thresholds. To evaluate the uncertainties of the product, we compared it with another satellite FT product also derived from passive microwave observations but at higher frequency: the resampled 37 GHz FT Earth Science Data Record
- (FT-ESDR). The assessment was carried out during the overlapping period between 2011 and 2014. Results show that 77.1% of their common grid cells have an agreement better than 80%. Their differences vary with land cover type (tundra, forest and open land) and freezing and thawing periods. The best agreement is obtained during the thawing transition and over forest areas, with differences between product mean freeze or thaw onsets of under 0.4 weeks. Over tundra, FT-AP tends to detect freeze onset 2–5 weeks earlier than FT-ESDR, likely due to FT sensitivity to the different frequencies used. Analysis with
- mean surface air temperature time series from six in situ meteorological stations shows that the main discrepancies between FT-AP and FT-ESDR are related to false frozen retrievals in summer for some regions with FT-AP. The Aquarius product is distributed by the U.S. National Snow and Ice Data Center (NSIDC) at https://nsidc.org/data/nsidc-0736/versions/1 with the DOI 10.5067/OV4R18NL3BQR.

### **1** Introduction

Seasonal freezing and thawing affect over half of the Northern Hemisphere. Landscape freeze/thaw (FT) state transitions show highly variable spatial and temporal patterns, with measurable influences to climate (IPCC, 2014; Peng et al., 2017; Poutou et al., 2004), hydrological (Gouttevin et al., 2012; Gray et al., 1985), ecological (Kumar et al., 2013; Black et al., 2000) and
biogeochemical processes (Selvann et al., 2016; Xu et al., 2013; Schaefer et al., 2011). The surface FT state affects the latent heat exchange and the energy balance at the interface between soil surface and the overlying medium. The vegetation growing season is sensitive to the annual non-frozen period (Kim et al., 2012), while vegetation net primary production and net ecosystemCO<sub>2</sub> exchange (NEE) with the atmosphere is impacted by FT timing variability (Barr et al., 2009; Kurganova et al., 2007). Comprehensive in situ observational long-term data sets for soil state characteristics across terrestrial environments are

10 still limited or inadequate, mostly for northern remote regions. Remote sensing in the thermal emission domain offers great potential for detecting changes in land-surface temperature, but is strongly limited by clouds, vegetation and snow cover (e.g. Langer et al., 2013). Spatially and temporally continuous information on soil freeze/thaw changes is lacking for the regions of both seasonal frozen ground and permafrost.

- Passive microwave remote sensing has been proven sensitive to the surface FT state due to large changes in surface dielectric properties between predominantly frozen and non-frozen conditions, and it offers global coverage. The remotely sensed FT detection capability at L-band (1.4 GHz) has been developed and validated in several studies (Zheng et al., 2017; Roy et al., 2017b; Rautiainen et al., 2012; Schwank et al., 2004). At L-band, the shallow depth contributing to the radiation (around 5 cm for an unfrozen soil) and the strong permittivity difference between water and ice (Δε<sub>ice/water</sub>) make it favourable for FT retrieval
- (Rautiainen et al., 2012; 2014). In recent years, passive L-band FT algorithms were created for NASA's Aquarius (Roy et al., 2015), ESA's soil moisture and ocean salinity (SMOS) (Rautiainen et al., 2016), and NASA's soil moisture active/passive (SMAP) (Derksen et al., 2017) missions. An FT Earth Science Data Record (FT-ESDR) was also produced using a higher microwave frequency at Ka-band (37 GHz) (Kim et al., 2017a). This product offers consistent and continuous global daily information on the FT state for several decades (1979-2016; Kim et al., 2017b). Observations were recorded by the scanning
- multi-channel microwave radiometer (SMMR), the special sensor microwave/imager (SSM/I), and the SSM/I Sounder (SSMIS).

This study presents the new Aquarius passive FT product for the Northern Hemisphere, distributed by the US National Snow and Ice Data Center (NSIDC) at nsidc.org/data/nsidc-0736/versions/1. The product precision and uncertainties are addressed

by comparing Aquarius FT retrievals with the FT-ESDR product for the overlapping period (2011 - 2014). The Aquarius passive FT product (referred to as FT-AP hereinafter) is based on the Aquarius weekly Level-3 L-Band brightness temperature (TB) product (Brucker et al., 2015; NSIDC: <u>http://nsidc.org/data/AQ3\_TB/versions/5</u>). The algorithm uses a relative frost factor (FFrel; see e.g. Rautiainen et al., 2014) based on normalized polarization ratio (NPR) temporal change detection (Roy

### Searth System Discussion Science Solutions Data

et al., 2015). To our knowledge, few intercomparisons between L- and Ka-band FT products exist (Derksen et al., 2017), and none evaluated inter-annual variability differences. However, it is well established that different frequencies interact differently with ground components (vegetation, soil, snow), canopy, etc. For instance, observations at L-band are less sensitive than at Ka-band to snow, plant biomass and surface roughness (Ulaby et al., 1986). Being less prone to disturbances above the ground, the L-band emission should give better information on the ground state in forested and snow-covered areas. In addition, since

- the L-band emission should give better information on the ground state in forested and snow-covered areas. In addition, since  $\Delta \epsilon_{ice/water}$  is larger at L-band ( $\Delta \epsilon_{ice/water} \approx 83$ ) than at Ka-band ( $\Delta \epsilon_{ice/water} \approx 10$ ) (Artemov and Volkov, 2014), there should be a higher sensitivity to the ground phase transition at L-band. Hence, because differences between products can be attributed to the microwave frequency and the algorithm used, the FT-AP is also compared with surface air temperature (SAT) observations.
- The main objective of this study is to present and evaluate the weekly FT-AP by comparing it to the FT-ESDR and to SAT observations across the Northern Hemisphere. First, we describe the new FT-AP product, designed by the algorithm developed by Roy et al. (2015), but applied across the Northern hemisphere. Then, we investigate the spatial and temporal FT variations from both FT-AP and FT-ESDR products over the Northern hemisphere. We then investigate the cause of the main differences between products from in situ information. The comparison aims to identify the similarities and differences between L-band

and Ka-Band FT products for further improvements of FT monitoring across the Northern hemisphere.

### 2. Method

### 2.1 Aquarius passive FT product (FT-AP)

The Aquarius FT product was generated using the Aquarius weekly averaged polar gridded L-band TB product distributed on the EASE-Grid 2.0, above the parallel 50° N, with a spatial resolution of 36 km x 36 km (Brucker et al., 2014). This formatted
TB was specially designed for the study of northern regions. The FT classification algorithm is based on a seasonal threshold approach (STA) using frost factor index (FFrel) (Eq. 1), introduced by Rautiainen et al. (2014), where FF<sub>NPR</sub> is the frost factor based on the normalized polarization ratio between TB at vertical and horizontal polarizations (TB<sub>V</sub> and TB<sub>H</sub>; Eq. 2). FF<sub>ff</sub> and FF<sub>th</sub> are reference frozen and thawed frost factors obtained respectively by averaging the five minimum and maximum FF<sub>NPR</sub> values of 2012 and 2013.

$$FFrel = \frac{FF_{NPR} - FF_{fr}}{FF_{th} - FF_{fr}}$$

$$FF_{NPR} = \frac{TB_V - TB_H}{TB_V + TB_H}$$
(1)
(2)

A threshold ( $\tau$ ) was determined by optimization to classify the surface as frozen or thawed if the FFrel is lower or higher than the threshold (Eq. 3).

 $\begin{array}{ll} \mathrm{If} & \mathrm{FFrel} < \tau \rightarrow \mbox{freeze} \\ \mathrm{or} \mbox{ if } & \mathrm{FFrel} > \tau \rightarrow \mbox{thaw} \end{array}$ 

(3)

The thresholds optimized in Roy et al. (2015) over Northern America for three basic land covers (tundra, forest, open land) were applied over the Northern Hemisphere using the Land Cover Classifications Derived from Boston University MODIS /

Terra Land Cover Data (LCC<sub>BU</sub>) (see Sect. 2.4).

Aquarius operated three non-scanning radiometers at different incidence angles  $(29.2^{\circ}, 38.4^{\circ} \text{ and } 46.3^{\circ})$  and with different 3 dB footprint sizes (respectively 76 km x 94 km, 84 km x 120 km and 97 km x 156 km). Based on the LCC<sub>BU</sub>, the thresholds found in Roy et al., (2015) were used to create FT maps for each radiometer. The three FT maps were then blended to create a fourth map, which offers more complete spatial coverage. For every grid cell, radiometer 2 (38.4°) is prioritized, while

- radiometer 3 is only used if data from the other radiometers is not available for the given grid cell. This blended algorithm was chosen based on the performance given for each radiometer in Roy et al. (2015) (radiometer 2 gave the best results, while radiometer 3 gave the worst results). Due to the width of Aquarius's wath and its revisit time, 16.5% of the terrestrial 36-km grid cells have less than 95% observations over the period and 16% were not measured at all. Thus, the intercomparison with the FT-ESDR product (Sect. 2.2) was only made when FT-AP data were available for a given date. The time span for this
- analysis runs from August 2011 with the first Aquarius observations to 31 December 2014 with the latest FT-ESDR data available at the time of our analysis.

#### 2.2 FT-ESDR product

The first version of the FT-ESDR product (Kim et al., 2011) was based on an STA similar to the FFrel but applied exclusively to the TB<sub>v</sub> at 37 GHz instead of the NPR. In the new extended product (Kim et al., 2017b; NSIDC: https://nsidc.org/data/nsidc-

20 0477/versions/4), a modified seasonal threshold algorithm (MSTA) was used to determine thresholds for each grid cell to obtain better accuracy. It consists of a grid-cell-wise weighted empirical linear regression relationship between the 37 GHz TB<sub>v</sub> measurements and daily surface air temperature estimates from the ERA-Interim global reanalysis.

The extended FT-ESDR product used in this study is derived from the SSM/I 37 GHz brightness temperatures (footprint of 38 km x 30 km) and resampled at a grid cell resolution of 25 km on the global Ease-Grid v1.0. The observations were recorded twice per day, which gives the possibility of attributing discrete frozen or thawed states for morning and afternoon. The final classification offers four discrete surface states: "frozen all day", "thawed all day", "frozen in AM and thawed in PM" (transitional) and "thawed in AM and frozen in PM" (inverse-transitional). In this study, the latter two classes were combined into a single transitional class. In order to compare the two products, the FT-ESDR was first spatially resampled to the EASE-

30 Grid 2.0 with the nearest neighbor method choosing the smallest distance between pixel centers. Then, FT-ESDR was temporally resampled based on the rule that the most frequently occurring class over the seven days of a week is adopted as

the value for the entire week. In cases where the frozen and thawed classes occurred with equal frequency during a single week (e.g. two days frozen, two days thawed and three days transitional), the transitional class was attributed. This latter class occurs mainly during the transition seasons of spring and fall. Thus, we assigned the transitional class to thawed class during spring and summer since it indicates the beginning of the thawing process and we assigned the transitional class to the frozen class

5 during fall and winter since it indicates the beginning of the freezing process. This FT-ESDR resampling procedure ensured that the two products were at the same temporal and spatial resolutions with only the frozen and thawed categories, making comparison possible.

### 2.3 Land cover classification

The land cover information (Fig. 1) comes from the EASE-Grid 2.0 LCC<sub>BU</sub> (Brodzik *and* Knowles, 2011; NSIDC:
nsidc.org/data/nsidc-0610/versions/1), using the same grid as the FT-AP product. The seventeen land cover classes were grouped to obtain four classes: tundra, forest, open land (savanna, cropland and grassland) and water (see Roy et al., 2015). Each grid cell was assigned its single most prominent class of land cover. All grid cells with more than 20% of water and ice indicated by the LCC<sub>BU</sub> were masked.