# Peer review of "Northern Hemisphere Surface Freeze/Thaw Product from Aquarius L-band Radiometers"

_Earth System Science Data, 2018_

## Referee Comment (RC1) · Anonymous Referee #1 · 8 Jun 2018

Monday, June 4, 2018 11:18 AM

The authors present a new FT product using Aquarius passive data and compare the results with the Ka-band FT-ESDR products. The results show some very interested global features over three years for three different landcovers. However, there is some lack of the algorithm description and discussion. The algorithm for this global F/T products is directly used the regional results from the Roy 2015 paper. I suggest, at least a few optimization over global region need to be tested and some basic screening should be applied to minimize the false alarm. The detail comments are as below,

1. In section 2.1, the author should give more descriptions of the general Aquarius mission, such as its native resolution for three radiometers, repeat cycle and etc. This info will help readers to better understand the what the FT-AP product represents. If

[Figure]

I understand right, the level 3 TB is a gridded products integrated both ascending and descending orbit in a weekly basis. The actual collecting time of the each pixel (in ∼100km Aquarius resolution) is still around 6am/6pm of the certain day but not representing the weekly average. While comparing to FT-ESDR, the weekly F/T from FT-ESDR is more of an averaged F/T status of the week. This instantaneous vs. weekly averaged comparison itself can lead up to a week of discrepancy. 2. Page 3 Line 25, why just generate reference from 2012 to 2013 instead of using the three year average? Is the five minimum/maximum value get from winter months and summer month or over the whole year? 3. It is worth listing out explicitly the thresholds that are used in generating the F/T map. Base on Roy 2015 paper, the thresholds were optimized over North American. How well is it applicable for the rest of the region? Especially in the latter session, the thawing process shows a great difference for the North American and Eurasia (Fig 5.) Is that related to the chosen of the threshold? 4. Figure (2b) wrong direction of the ] in the figure label 5. In section 3.3, when comparing with the in situ weather stations, the surface air temperature is definitely an important indicator of the soil freeze/thaw. However, it's still an indirect way of predict F/T status of the soil. Although the soil moisture and soil temperature may also have some ambiguity to determine the F/T, it has more direct information of the soil itself and should be included in the discussion when they are available.

---

## Referee Comment (RC2) · Anonymous Referee #2 · 10 Oct 2018

The authors present a new database of freezing/thawing state of the surface in the Northern High Latitudes. This product is derived from the satellite mission Aquarius. This database is of interest and is complementary to what already exists. Detecting soil state is difficult especially during the transition periods. Using remote sensing data acquired at various frequencies can help to better monitor these soil conditions.

My main comment concerns the definition of the thresholds. There are computed by Roy et al., and applied globally.

Detail commentsÂă:

Section 2.1Âă: More details on the Aquarius mission are needed, such as orbits used (ascending, descending, both?)

[Figure]

It would be interesting to discuss more the thresholds, the values and how they are computed.

Page 4, lines 8 to 14, what about the 1st radiometerÂă? The authors mention that their method uses radiometer 2 and 3, and radiometer 1Âă?

Page 14, paragraph 2.3Âă: What about pixels that are heterogeneous, e.g. having one class covering 51Âă% of the surface and another one 49Âă%. Is the threshold of the main class adapted to these casesÂă?

ÂńÂăwhich difficultiesÂăÂż. It is not clear what the authors mean by difficulties. It is to be clarified and discussed.

Page 9, 3rd line : ÂńÂăobvisous false retrievalsÂăÂż. Please clarify, why these false retrievals are obviousÂă?

Page 10, 4th lineÂă: ÂńÂăhorizontal shiftÂăÂżÂă. Do the authors refer to the shift in FallÂă?

Figures 4 and 5 : Please add the name of database and the year along with the indices a) b) ... it would help the readers.

Figure 6Âă: it is not convenient to have the figures on several pages, but it would definitely help if the authors could add the legend and the name of the station on each figures. Actually the legend is on page 17 whereas the name of the stations are described on page 21, which is a bit annoying.

Figure 6Âă. It seems the time series depicted by bullets are not continuous. There are missing points, what happens there if the conditions are neither Thawed nor frozen.

The authors discuss in many occasions the use of NPR (its dynamic, seasonal range ..) and thresholds but it is difficult for a reader to evaluate their comments without illustrations to support their comments. Figures showing time series of NPR and thresholds are needed to help the discussion.

Rowlandson et al. Is under review. Please update or remove if not accepted.

AbstractÂă: Ration ⇒ ratioÂă?

---

## Author Comment (AC1) · 6 Nov 2018

**Revision of Manuscript Number:** essd-2018-25

In blue: Reviewer's comments.
R1 or R2 = first or second reviewer respectively.
C# = comment number when indicated

In black: Answers to reviewer
(Page): the pages where modification to text were added
In black and italic: Modification added to text.

**Comments from the Reviewers:**

R1-C1.1: In section 2.1, the author should give more descriptions of the general Aquarius mission, such as its native resolution for three radiometers, repeat cycle and etc. This info will help readers to better understand the what the FT-AP product represents

R2: Section 2.1: More details on the Aquarius mission are needed, such as orbits used (ascending, descending, both?).

The native resolutions were already given in section 2.1 (the FT-AP description). Details on the TB value are added, in order to explain that the TBs are average based on every measurement available during a given week, combining ascending and descending orbits. The reader is referred to the detailed description of the Aquarius dataset explained by Brucker et al., 2015.

*p.3: For each Aquarius radiometers, the product average TB values calculated from every measurement made during a week, combining ascending and descending orbits.*

R1-C1.2: If I understand right, the level 3 TB is a gridded products integrated both ascending and descending orbit in a weekly basis. The actual collecting time of the each pixel (in ~100km Aquarius resolution) is still around 6am/6pm of the certain day but not representing the weekly average. While comparing to FT-ESDR, the weekly F/T from FT-ESDR is more of an averaged F/T status of the week. This instantaneous vs. weekly averaged comparison itself can lead up to a week of discrepancy.

On the one hand, the temporal resampling of the FT-ESDR product uses only the seven values of a given week (one every day) to create a single class representing the whole week, according to the classification method described in the article (p.4-5). On the other hand, FT-AP uses the Aquarius weekly averaged TB, which results from all the TB values of a given week, for a given pixel. Since we resample the FT-ESDR on a weekly basis, there is no discrepancy in the comparison because both products are on the same weekly basis. Description added at the p.3 concerning Aquarius weekly TB value

(mentioned in the previous comment) should help to avoid that confusion. Moreover, an adjustment has been made in the description of the temporal resampling method at p.5 L10:

*p.5 L10: Then, FT-ESDR was temporally resampled at the same weekly calendar than the FT-AP. The temporal FT-ESDR sampling procedure was based [...]*

R1-C2: Page 3 Line 25, why just generate reference from 2012 to 2013 instead of using the three year average? Is the five minimum/maximum value get from winter months and summer month or over the whole year?

The sentence has been adjusted for a better description of the technique. So, it was not only from 2012 and 2013, but indeed over the whole series. Also, $FF_{fr}$ and $FF_{th}$ were calculated for every pixel.

*p.3: $FF_{fr}$ and $FF_{th}$ are reference frozen and thawed frost factors obtained for each pixel and each radiometer by averaging, respectively, the five minimum $FF_{NPR}$ found during winters (January and February) and five maximum $FF_{NPR}$ found during summer (July and August) over the three available dataset period.*

R1-C3: It is worth listing out explicitly the thresholds that are used in generating the F/T map. Base on Roy 2015 paper, the thresholds were optimized over North American. How well is it applicable for the rest of the region? Especially in the latter session, the thawing process shows a great difference for the North American and Eurasia (Fig 5.) Is that related to the chosen of the threshold?

R2: It would be interesting to discuss more the thresholds. the values and how they are computed.

Both reviewers had a similar comment, they are addressed together here. A table with the threshold values has been added. More details, based on Roy et al. (2015) results, were added, showing that the optimization technique only slightly improved the accuracy. According to these results, we do not think it is an issue to use them over the northern hemisphere. For tundra site, still based on Roy et al. (2015) results, a broad range of threshold values ([0.3-0.7]) caused an insignificant variation of accuracy. Given that, we do not think the thresholds are a source of discrepancies between the products. Moreover, in order to illustrate the effectiveness of the used thresholds applied on extended circumpolar areas, we added, in Figure 6, NPR time series with the corresponding thresholds for all the studied sites (discussed in a next comment below).

*p.4: The optimization method calculates the threshold that gives the best accuracy when the product retrievals is compared to in situ air temperature stations. It was shown that*

*optimized thresholds only slightly improved the accuracies by 1% to 4% compared to a fixed threshold of 0.5. For tundra site, a broad range of threshold values ([0.3-0.7]) caused an insignificant variation of accuracy.*

*p.4: Table 1: Thresholds (τ) applied in Eq. 3 for the whole circumpolar area, derived from the Roy et al. (2015)*

| Beam | Tundra | Forest | Open land |
|------|--------|--------|-----------|
| 1 | 0.41 | 0.46 | 0.31 |
| 2 | 0.69 | 0.55 | 0.31 |
| 3 | 0.63 | 0.54 | 0.41 |

We decided to delete one sentence p.13 because it was not supported by any statistical values. The conclusion is rather subjective. More precisely, differences between products in North America and in Eurasia are clearly more pronounced for the thawing season in 2014, but it is not trivial to make a clear conclusion for 2012 and 2013 seasons.

Sentence deleted: p.13: *More specifically, FT-ESDR tends to retrieve thaw earlier in North America, while FT-AP retrieves thaw earlier in Eurasia. These differences are more pronounced in 2014 (Fig. 5c).*

R1-C4: Figure (2b) wrong direction of the ] in the figure label

It is a common way in statistics to indicate which numbers on the limits of a group are included or excluded of the group. So, we decided to keep this annotation.

R1-C5: In section 3.3. when comparing with the in situ weather stations. the surface air temperature is definitely an important indicator of the soil freeze/thaw. However. it's still an indirect way of predict F/T status of the soil. Although the soil moisture and soil temperature may also have some ambiguity to determine the F/T. it has more direct information of the soil itself and should be included in the discussion when they are available.

Along with surface air temperature datasets, only precipitation datasets were available with the selected stations. Our recent studies showed that it is not trivial to link or validate L-band satellite TB measurements and FT retrievals to soil temperature, soil moisture and precipitation. It is common in the FT satellite products to use surface air temperature as the in situ reference (Kim et al., 2017; 2011; Derksen et al., 2017; Roy et al., 2015).

In order to justify our choice of in situ reference, we already put this sentence at p.16:

p.16: *SAT was chosen as in situ reference since Roy et al. 2015 showed that SAT was the best proxy to validate satellite FT products.*

We modified the comment in the Discussion Section to outline the possible effect of soil moisture in the retrievals:

*p.23: While SAT is an indirect way to derive information on soil FT state, it was used in this study because it is a more homogenous reference than soil temperature. Soil temperature influences the emission (by Planck's law) of landscape elements such as soil, snow and vegetation. Moreover, L-band TB are also sensitive to soil moisture (see the review from Wigneron et al., 2017) which could have strong spatial variability at local scale. Microwave emissions detected by a satellite radiometer with all the spatial variability of the environment within a pixel cannot be solely validated by SAT, since it does not consider phenomena like thermal inertia and latent heat exchange.*

Added reference:
*Wigneron, J., Jackson, T. J., Neill, P. O., Lannoy, G. De, Rosnay, P. De, Walker, J. P., Ferrazzoli, P., Mironov, V., Bircher, S., Grant, J. P., Kurum, M., Schwank, M., Munoz-sabater, J., Das, N., Royer, A., Al-yaari, A., Bitar, A. Al, Fernandez-moran, R., Lawrence, H., Mialon, A., Parrens, M., Richaume, P., Delwart, S. and Kerr, Y.: Modelling the passive microwave signature from land surfaces : A review of recent results and application to the L-band SMOS & SMAP soil moisture retrieval algorithms, Remote Sens. Environ., 192(January), 238–262, doi:10.1016/j.rse.2017.01.024, 2017.*

R2: Page 4. lines 8 to 14. what about the 1st radiometer? The authors mention that their method uses radiometer 2 and 3. and radiometer 1?

We put the information explicitly in the same sentence to avoid any confusion.

*p.4: For every grid cell, radiometer 2 (38.4°) was prioritized, then radiometer 1 (29.2°) was used, while radiometer 3 was only used if data from the other radiometers were not available for the given grid cell.*

R2: Page 5. paragraph 2.3: What about pixels that are heterogeneous. e.g. having one class covering 51% of the surface and another one 49%. Is the threshold of the main class adapted to these cases?

There is no distinction between homogenous and heterogeneous pixels in this analysis and there was no threshold adjustment. We applied exactly the technique used in Roy et al. (2015) when the principles for the database was developed. The following information was added to avoid any question or confusion.

*p.5: Each grid cell was assigned its single most prominent class of land cover which is used for the selection its thresholds (Table.1).*

R2: Page7: "Which difficulties". It is not clear what the authors mean by difficulties. It is to be clarified and discussed.

We replaced "difficulties" by "lower accuracies", referring to the comparison between satellite retrievals and in situ measurements made the studies mentioned in that same sentence.

p.7: *Those lower percentages correspond to regions where lower accuracies to detect the FT were already noted in Roy et al. (2015) and Kim et al. (2017a) (see Sect. 4).*

R2: Page 9. 3rd line: An obvious false retrievals. Please clarify. why these false retrievals are obvious?

The obvious false retrievals are thoroughly addressed in a later section of the paper. Since it is the first mention in the text about the obvious false retrieval, we understand the question and the confusion. So, the sentence has been adjusted:

p.9: *To reduce the effect of obvious false frozen retrievals in summer (discussed below) on the analysis and to focus on the differences primarily related to the physics of the measurements [...]*

R2: Page 10. 4th line: A horizontal shift. Do the authors refer to the shift in Fall?

The mention of "in fall" has been added at the beginning of the sentence describing the horizontal shift, because the analysis is indeed focus on fall freezing periods.

*p.10: In fall, the horizontal shift between the curves indicates time delays (Δtime) for the two products to reach the same percentage of frozen grid cells.*

R2: Figures 4 and 5: Please add the name of database and the year along with the indices a). b) ... it would help the readers.

Names of database and the year were added to the figures. The name "diff." was used for the bottom figure instead of difference or "FT-AP minus FT-ESDR" to not overload the figure. Captions were adjusted to include the definition of "diff." as follows:

*p.13: Figure 4: Freeze onset maps, where colors indicate the week of year, for a) 2011, b) 2012, c) 2013 and d) 2014 with FT-AP (top), FT-ESDR (middle) and difference between the products (Diff. = FT-AP minus FT-ESDR; bottom)*

*p.16: Figure 5: Thaw onset maps, where colors indicate the week of year, for a) 2012, b) 2013 and c) 2013 with FT-AP (top), FT-ESDR (middle) and difference between the products (Diff. = FT-AP minus FT-ESDR; bottom)*

R2: Figure 6: it is not convenient to have the figures on several pages. but it would definitely help if the authors could add the legend and the name of the station on each figures. Actually the legend is on page 17 whereas the name of the stations are described on page 21. which is a bit annoying.

The name and a legend have been added to each figure. They all have been regrouped one after the other to facilitate the reading.

R2: Figure 6. It seems the time series depicted by bullets are not continuous. There are missing points. what happens there if the conditions are neither Thawed nor frozen.

Missing points happen when none of Aquarius radiometers had collected a measurement during a specific week. In p.4, a sentence mentions that this possibility can occur. To avoid any question or confusion, some clarifications have been added in p.16 before the presentation of the figures.

*p.16: Discontinuities in the series (Fig.6a-f) is caused by the absence of Aquarius observations in a given week.*

R2: The authors discuss in many occasions the use of NPR (its dynamic. seasonal range ..) and thresholds but it is difficult for a reader to evaluate their comments without illustrations to support their comments. Figures showing time series of NPR and thresholds are needed to help the discussion.

NPR series have been added to every figure with their thresholds values. The legend of Figure 6 has been updated to give the information about the new series

*p.21: Figure 6. FT detection for each reference site (see Table 1), with FT-ESDR (red dots) and FT-AP (blue dots) against surface air temperature (black dots and blue line) in a) Kamchatka, b) Quebec, c) Alaska, d) Siberia, e) Kazakhstan and f) Saskatchewan.*

*NPR series (top) contain the combination of available Aquarius observations following the prioritization of radiometer 2, radiometer 1 and then radiometer 3 (sect. 2.1). NPR threshold values (blue dot) according to Eq.1 with the corresponding beam number showed on the right.*

R2: Rowlandson et al. Is under review. Please update or remove if not accepted.

The reference is now up to date

*p.23: Rowlandson et al. (2018)*

*p.28: Rowlandson. T.. A. Berg. A.. Roy. A.. Kim. E.. Pardo Lara. R.. Powers. J.. Lewis. K.. Houser. P.. McDonald. K.. Toose. P.. Wu. A.. De Marco. E.. Derksen. C.. Entin. J.. Colliander. A. and Xu Xiaolan: Capturing Agricultural Soil Freeze/Thaw State through Remote Sensing and Ground Observations: A Soil Freeze/Thaw Validation Campaign. Remote Sens. Environ.. 211. 59-70. doi:10.1016/j.rse.2018.04.003. 2018.*

R2: Abstract: Ration / ratio?

Corrected